# The Effect of Y^3+^ Addition on Morphology, Structure, and Electrical Properties of Yttria-Stabilized Tetragonal Zirconia Dental Materials

**DOI:** 10.3390/ma15051800

**Published:** 2022-02-27

**Authors:** Marko Jakovac, Teodoro Klaser, Arijeta Bafti, Željko Skoko, Luka Pavić, Mark Žic

**Affiliations:** 1Department of Fixed Prosthodontics, School of Dental Medicine, University of Zagreb, Gundulićeva 5, 10000 Zagreb, Croatia; jakovac@sfzg.hr; 2Ruđer Bošković Institute, P.O. Box 180, 10000 Zagreb, Croatia; teodoro.klaser@irb.hr (T.K.); lpavic@irb.hr (L.P.); 3Faculty of Chemical Engineering and Technology, University of Zagreb, Marulićev trg 19, 10000 Zagreb, Croatia; abafti@fkit.hr; 4Department of Physics, Faculty of Science, University of Zagreb, Bijenička, c. 32, 10000 Zagreb, Croatia; zskoko@phy.hr

**Keywords:** zirconia, PXRD, impedance, EEC, conductivity, dental material

## Abstract

Zirconia (ZrO_2_), a widely known material with an extensive range of applications, is especially suitable for dental applications. This kind of dental material is produced in the form of blocks or discs (mostly discs—depending on CAD/CAM machines) by cold isostatic pressing (CIP). Such discs are subsequently milled by CAM/CAD technology into a desirable form. Due to the application of CIP, the resulting discs consist of different yttria-stabilized tetragonal zirconia polycrystal (Y-TZP) powders, additives and pigments. The diverse composition of the discs (e.g., different Y^3+^ content) governs material properties, such as hardness, toughness and translucence. The aim of this work was to investigate the impact of Y^3+^ addition on the grains and grain boundaries, on the ZrO_2_ phases lattice parameter values and on the electrical equivalent circuit parameters of the prepared Y-TZP samples. The disc-shaped samples were prepared by using CAM/CAD technology. It was observed that the grain size and the grain density were increased by Y^3+^ addition. The sample with the lowest Y^3+^ content was characterized by the highest portion of the tetragonal phase, whilst the disc with the highest Y^3+^ addition consisted mainly of the cubic phase. It was also observed that at the higher Y^3+^ ion concentrations, these ions mainly incorporated the tetragonal phase. Furthermore, conductivity investigations showed that the resistivity of the grains in the samples with the higher Y^3+^ concentrations was decreased as these ions were mainly trapped in the grain boundary. On the other hand, the Y^3+^ trapping increased the capacitance of the grain boundary.

## 1. Introduction

Zirconia (ZrO_2_) is a well-known stable material with different ranges of applications, varying from clinical applications [1] and dental materials [2] to fuel cells [3]. Therefore, it has been a focus of numerous scientific papers that involve both in vitro [4] and in vivo [5] studies. ZrO_2_ is accessible and attractive for dental applications [2] due to the fact that it can be processed by computer-aided manufacturing (CAD/CAM) technology [2]. In addition, there is a wide range of different dental ceramic materials with clinical applications [6] that can be shaped by CAD/CAM technology. The application of this technology facilitates the development and investigations of ZrO_2_ as it is possible to both shape and prepare this material in numerous ways [7].

Previous studies showed [8,9] that some aspects of the electrical and structural properties of dental materials (including zirconia) have not been thoroughly tested yet. It is known that Y^3+^ stabilizes the tetragonal form at room temperature and it boosts zirconia’s mechanical properties [2]. However, the impact of higher Y^3+^ content on both the phase composition and lattice parameter values of new yttria-stabilized tetragonal zirconia polycrystal (Y-TZP) materials should be constantly revisited. After all, by using CAD/CAM technology, samples and experiments can be more easily designed.

It is a well-known fact that the properties and structure of zirconia have been thoroughly tested [10,11,12]; however, having said that, the impact of different amounts of Y^3+^ on the electrical properties has not been fully illuminated for new dental Y-TZP materials yet. The amount of Y^3+^ is also responsible for both the oxygen vacancies amount and the oxygen ion conductivity [3]. Although there is a myriad of studies that have addressed, e.g., the impact of sandblasting on ZrO_2_ [8,13,14,15], it seems that Impedance Spectroscopy (IS) has not yet been fully exploited when dealing with the dental Y-TZP material(s).

The electrical properties of ZrO_2_-based materials can be more rigorously examined when using the electrical equivalent circuit (EEC) model as a diagnostic tool [16,17,18,19]. For example, by studying EEC parameter trend(s), it should be possible to gain a deeper insight related to the location of the Y^3+^ ions within the material. However, one should be aware of the fact that nowadays, an EEC model-free approach to analyze IS data is gaining prominence [20,21,22]. Nevertheless, EEC analysis still offers a fair amount of data; thus, it should be used more frequently while investigating this sort of material (see, e.g., [8]). For example, by using EECs, it is possible to monitor intra-grain and inter-grain resistivity [23,24].

These sorts of studies are also significant because they can point out defects in the material, e.g., holes, cracks and chipping [8,25] that could be outcomes of the CAM/CAD processing. Therefore, it is of common interest to study the impact of processing (e.g., milling) on new dental materials with the intention to predict clinical implications such as an increased risk of fracture or accelerated aging. Y-TZP materials are stabilized at room temperature by Y^3+^ addition [26,27] but these materials can be easily transformed from the tetragonal to the monoclinic phase [28], even with the sandblasting energy. This transformation is currently well investigated as its impact on the volume and stability of Y-TZP is substantial.

Herein, we intend to investigate several different Y-TZP samples with various amounts of Y^3+^. Y^3+^ addition has several vital implications for clinical restoration; hence, we tested some of the material(s) properties, such as structural, morphological and electrical properties. Above all, the impact of Y^3+^ on the surface morphology and structure was thoroughly inspected by scanning electron microscopy (SEM) and Powder X-ray Diffraction (PXRD).

This study aimed to investigate several discs with different Y^3+^ content. One of the main aims was to observe the impact of Y^3+^ on the ZrO_2_ phase composition and on the lattice parameter values. The study also focused on the impact of Y^3+^ on the grain size and grain boundaries, and on the EEC parameter values. The fact that distinguishes this study from the existing ones is the application of IS as it revealed additional insights related to the impact of Y^3+^ ions on the electrical properties of the zirconia-based dental materials.

## 2. Materials and Methods

### 2.1. Sample Preparation

Yttria-stabilized tetragonal zirconia polycrystal (Y-TZP) discs with gradually ascending Y^3+^ content were prepared from one IPS e.max^®^ ZirCAD Prime block/disc (Ivoclar Vivadent, Shaan, Liechtenstein) (Figure 1). The discs, with a thickness of 1 mm and a diameter of 10 mm (Table 1), were prepared in an ordinary dental laboratory. Preparation was conducted by utilizing a dental milling device (Cerec, MCX 5, Dentsply Sirona, Bensheim, Germany).

Then, the discs were sintered at 1500 °C by using a heating ramp recommended by the manufacturer. The samples were prepared without post-processing, i.e., there were no additional surface treatments. It was decided that the disc shape be used since this specific profile can be analyzed by all the experimental techniques utilized in this study.

The sample with the lowest Y^3+^ content is named S1 (3Y-TZP; 3 mol% Y_2_O_3_), while the specimen with the highest Y^3+^ addition is designated as S9 (5Y-TZP; 5 mol% Y_2_O_3_). The exact chemical composition of the Y-TZP block (Figure 1) applied in this work is not known, but according to the manufacturer documentation, it is mainly composed of ZrO_2_, Y_2_O_3_, HfO_2_, Al_2_O_3_ and other oxides, additives and pigments. However, the trend in the Y^3+^ added in the prepared discs is known (Table 1) due to the applied milling procedure.

### 2.2. Method for Surface Investigation

The prepared Y-TZP samples’ morphologies were investigated by scanning electron microscopy (SEM). Herein, SEM images were collected by a thermal field emission scanning microscope (FE SEM, model JSM-7000F, JEOL Ltd., Akishima, Japan) operating at 3 kV.

### 2.3. Method for Structural Identification

Powder X-ray Diffraction (PXRD) data were collected by applying a Bruker Discover D8 diffractometer (Karlsruhe, Germany) supplied with a LYNXEYE XE-T detector. Measurements were taken in Bragg–Brentano geometry (1D) applying CuKα radiation (1.54 Å). The angular range 2θ varied from 10 to 70. The data were obtained by using a step size of 0.02 and measuring time of 27 s/step. Rietveld structure refinement was conducted by using the HighScore Xpert Plus program 3.0 (Malvern Panalytical, Almelo, The Netherlands).

### 2.4. Electrical Properties

The SC7620 sputter coater (Quorum Technologies Ltd., Laughton, East Sussex, UK) was utilized to deposit gold electrodes (7 mm in diameter). The gold was sputtered onto both sides of the sample discs. The impedance response was recorded by applying an impedance analyzer (Novocontrol Alpha-AN dielectric spectrometer, Novocontrol Technologies GmbH&Co. KG, Hundsangen, Germany). The frequency range was in the 0.01 Hz–1 MHz interval and the applied temperature range varied from 303 to 483 K (±0.5 K).

## 3. Results and Discussion

### 3.1. Morphology Investigation

The morphology of the investigated S1–S9 samples is presented in Figure 1. These samples show typical morphology of the sintered yttria-stabilized tetragonal zirconia polycrystal (Y-TZP) materials (i.e., grains and grain boundaries). The observed morphology indicates that these samples were not sandblasted, which usually precedes the dental application. To clarify, the crowns are generally sandblasted in the dental laboratory for cleaning purposes. In addition, the sandblasting damages the specific Y-TZP morphology [25]; thus, the sandblasted samples do not show grains and grain boundaries.

Figure 1 clearly suggests that the grain size of the discs is gradually increasing with Y^3+^ addition, i.e., S9 has the highest Y^3+^ content and the biggest grain size. A higher Y^3+^ content would further increase the grain size, which in turn could compromise the material stability. However, nowadays there have been improvements in Y-TZP production technology which resulted in dental materials with an even higher Y^3+^ (>5 mol%) content.

The sample S1 (Figure 2) demonstrates a less dense morphology in comparison to the sample S9 (Figure 3). Although samples S1–S9 were prepared (i.e., sintered) by using the same temperature heating ramp, the effect of sintering on the morphology and density is different in this study. To elaborate, these materials are fabricated by cold isostatic pressing (CIP) [29], which is a method for preparing compact powder materials. However, these powder materials may consist of various additives to ensure better densification during the sintering process.

The densification of the CIP-processed Y-TZP materials can be promoted by using densification additives [30]. These kinds of materials assist densification through liquid sintering, which in turn facilitates ion diffusion. The diffusion of Y^3+^ within the CIP-processed Y-TZP material during the sintering stage is inevitable, which can promote the grain(s) growth and the phase transformation(s). It should be mentioned that densification can be conducted by using different sintering techniques [31], but densification without an additional grain growth is one of the main tasks during the sintering process [32].

Next, the less dense morphology of the sample S1 can be explained as follows. Firstly, the added 3Y-TZP powder (prior to sintering) is tougher and harder to compress (Figure 2), and secondly, the quantity of the densification promoters may be lower [30]. On the other hand, the morphology of the sample S9 is denser (Figure 3) as 5Y-TZP powder is less hard and there may be more additives for densification. The Y-TZP materials studied in this work contain diverse compounds and additives that are not specified by the manufacturer. Thus, at this point of the study, it is hard to distinguish the impact of densification additives, ion diffusion during the sintering process, and the Y-TZP powders’ hardness on the morphology and density of the samples S1–S9.

The impact of the milling process can be observed in sample S1 as there is damage (i.e., voids) in the material’s surface (Figure 2). On the other hand, such defects are not detected in sample S9 (Figure 3). The lack of imperfections in the sample S9 can be explained by a more dense morphology that increased the material’s toughness. This boosted toughness may be assigned to the presence of the densification additives and/or promoted ion diffusion during the sintering process.

Please note that defects, such as in the sample S1, can be easily removed by surface processing such as, e.g., sandblasting which is accompanied by both the phase transformation and volume ingress (3–5%). Therefore, this transformation can “seal up” defects at the surface (see, e.g., [8]) although this transformation decreases the material’s fractural resistance. These imperfections can act as centers for aging, and for this reason, the material is commonly sandblasted. However, it should be mentioned that there are different approaches for surface processing, for instance, laser treatment [33], which is not usual in dental practice.

### 3.2. Structural Investigation

Since zirconia-based powder materials are well crystalline, Powder X-ray Diffraction (PXRD) is a perfect tool for studying both the crystallinity and structure of ZrO_2_-based materials [34,35,36]. The PXRD patterns of the S1–S9 samples are presented in Figure 4 and they show the presence of the tetragonal and cubic phases identified by the Inorganic Crystal Structure Database (ICSD) card no. 164862-ICSD and card no. 185125-ICSD. The tetragonal zirconia phase at room temperature is stabilized by the Y^3+^ ions [2]. Figure 4 shows that an increased Y^3+^ content (see Table 1) yields a higher portion of the cubic phase. To rephrase, the cubic phase portion is lowest in the case of the sample S1 and highest in the case of the sample S9. However, to offer a more precise PXRD study, the fraction of both phases was determined by using the Rietveld refinement method, and the results are given in Table 2 and Figure 4 (right).

Table 2 shows that Y^3+^ is responsible for the formation of the cubic phase in the investigated S1–S9 Y-TZP samples. Samples S1–S5 have the lowest Y^3+^ content and the highest portion (>70%) of the tetragonal phase. It should also be mentioned that the tetragonal phase lattice parameters (*a* and *c*) in samples S1–S5 do not show any specific trend (Table 2). As these Y-TZP materials are prepared by the CIP method (i.e., several types of powders are added before the pressing step [29]), there might be some imperfections. In this study, these imperfections can be observed in the lack of the phase composition trend in the samples S1–S5, which may be responsible for the similar morphology (Figure 1).

Furthermore, as the Y^3+^ content is heightened, the fraction of the cubic phase is enlarged, which decreases the hardness and toughness of the material, but at the same time, it also increases translucence. The Y-TZP samples in this work are partially cubic (22–70%) which is a finding that corresponds well to the literature [37]. To be more specific, the cubic fraction amount (22%) in 3Y-TZP (sample S1) agrees well with the cubic fraction quantity (23.2%) in the 2.4Y-TZP sample investigated in [38]. The phase composition is of vital importance for dental applications and restorations, and according to Table 2, the composition is controlled by the Y^3+^ content. On the other hand, the grain size in the samples S6–S9 gradually increases and, in parallel, the tetragonal phase lattice parameter *a* shows an ascending trend (Table 2). This ascending trend suggests that when Y^3+^ addition is higher, Y^3+^ mainly incorporates the tetragonal structure.

Next, sample S9 (see Figure 1 and Figure 3) is characterized by two different grain size distributions. The origin of these size distributions can be assigned to the fact that the manufacturer produces the Y-TZP block (Figure 1) by adding two different powders (3Y-TZP and 5Y-TZP) prior to the CIP pressing step. However, the impact of diverse densification additives on the grain growth process should also be considered [32]. According to the SEM (Figure 1) and PXRD (Figure 4 and Table 2) studies, it is safe to say that the smaller grains are mostly composed of the tetragonal phase, whilst the bigger grains are mainly defined by the cubic phase.

### 3.3. Electrical Investigation

Y-TZP materials are oxygen ion conductive electrolytes [39,40] and they also have a wide application in, e.g., solid oxide fuel cells (SOFC) [23,41]. These materials are applied in SOFC due to their high chemical (and electrical) stability and low costs [41], which are properties that are essential for dental applications. Furthermore, SOFC performances highly rely on both the preparation method and porosity of the Y-TZP materials which can also be studied by Impedance Spectroscopy (IS) [42].

The total electrical resistivity of most of the ceramic electrolyte systems (e.g., Y-TZP), can be assigned to both the migration of the oxygen ions within the grains (i.e., intra-grain) and to the blocking of the migrating oxygen ions at grain boundaries (i.e., inter-grain) [39]. Thus, the complex impedance spectra of polycrystalline solid ion conductors (e.g., Y-TZP) are usually characterized by at least three depressed semicircles (i.e., arcs) that can be assigned to grains and grain boundaries, and polarization at the electrode interface [43]. However, these arcs might not all be observable at high temperatures [43].

Figure 5a shows the conductivity spectra of the investigated S1–S9 discs and there are several details that should be commented on. First, the impedance response of Y-TZP samples is governed by the content of oxygen vacancies which is controlled by the Y^3+^ concentration. Second, all samples have diverse behavior in the low-frequency region, which is governed by electrical polarization at the electrode surface. Third, the “mid” frequency region is defined by the blocking of the migrating oxygen ions at grain boundaries, and the conductivity in this region increases by Y^3+^ addition. This suggests that some Y^3+^ ions were unable to diffuse into the grains (bulk) during the sintering process. Forth, the impedance response in the high-frequency region is controlled by the migration of the oxygen ions within the grains (Figure 5a). It appears that the conductivity in this region is decreasing with Y^3+^ content, which also suggests that these ions may be “trapped” in the grain boundary during the sintering process. Finally, the total DC conductivity of the samples does not show any specific trend. This observation may be explained by the fact that these samples consist of different additives and pigments that prevent the formation of a conductivity trend.

Nowadays, one popular approach to investigate impedance spectra is to apply an Equivalent Electrical Circuit (EEC) [42]. Firstly, it is important to choose an EEC that mimics the physical behavior of the system under investigation. However, this might be a daunting task, as one semi-depressed semicircle can be formed by several ones. Nevertheless, there are model-free approaches such as the Distribution Function of Relaxation Times [20,21] that can be applied to extract the number of electrical/electrochemical processes in the system under investigation. 

In this study, an EEC consisting of two parallel circuits was applied as each circuit can be assigned to one physical process that occurs within the samples (Figure 6). The first R_1_Q_1_ parallel circuit reflects the oxygen ion migration within the grains (i.e., intra-grain), and is characterized by a resistor (*R*_1_) and a constant-phase element (*Q*_1_). The second R_2_Q_2_ parallel circuit imitates blocking of the migrating oxygen ions through the grain boundaries. Finally, at the end, the electrode polarization at the sample surfaces is modeled by *Q*_el_.

The EEC displayed in Figure 6 can also be represented in the form of the transfer function:(1)$$Z\left(\omega \right)=\frac{1}{1/R_{1}+{\left(i\omega Y_{0,1}\right)}^{n_{1}}{}_{}}+\frac{1}{1/R_{2}+{\left(i\omega Y_{0,2}\right)}^{n_{2}}{}_{}}+{\left(i\omega Y_{0,el}\right)}^{n_{3}}{,}_{}\;$$
where *R*, *i*, *ω* and (*iωY_0_*)*^n^* are the resistance, the imaginary unit, the angular frequency (2*πf*) and the admittance of the constant phase element. The impedance data of the S1–S9 samples were fitted by using (1) and the Levenberg–Marquardt algorithm [44,45], and the extracted EEC parameters are displayed in Table 3.

At first sight the values in Table 3 do not show any specific trend, which can be assigned to the diversity in the grain size, porosity (Figure 1), phase content (Table 2), additives and Y^3+^ addition (Table 1). However, a quick analysis of Table 3 suggests that the *R*_1_ and *Y*_0,2_ values show the same trend as the ascending trend of both the cubic phase fraction and Y^3+^ addition. Since the *R*_1_, *Y*_0,2_ and cubic fraction values differ by several orders of magnitude, the corresponding values were normalized to fall in the range between 0 and 1 by the following formulation:(2)$$x_{normalized}=\frac{x-x_{minimum}}{x_{maximum}-x_{minimum}},$$
and the normalized values are given in Figure 7.

Figure 7 shows the ascending trend of *R*_1_ (i.e., intra-grain resistivity) values obtained for samples S4–S9. The ascending *R*_1_ trend is accompanied by the increasing Y^3+^ ion content in the S4–S9 samples (Table 1), which suggests that the Y^3+^ content within the grains is lower. To elaborate, according to Figure 1, the samples with a higher Y^3+^ content are denser, which suggests that a more compact grain boundary was formed. As grain boundary ion diffusion is the dominant densification mechanism [46], Y^3+^ ions may be trapped within this boundary which additionally decreased the grain conductivity (Figure 5a).

The second ascending trend is detected for the *Y*_0,2_ values of the samples S5–S9 (Figure 7). Since the corresponding *n*_2_ values are > 0.8 (Table 3), *Y*_0,2_ can be commented on in terms of pseudocapacitance. According to Figure 7, the ascending trend of the *Y*_0,2_ values is also accompanied by heightened Y^3+^ content. It is fair to say that as more Y^3+^ ions are trapped within the grain boundary, the higher oxygen vacancies content increases both the pseudocapacitance and *Y*_0,2_ values.

As the conductivity of the Y-TZP samples is governed by the portion of Y^3+^ ions, it was decided that the activation energy (*E_dc_*) for DC conduction be determined. *E_dc_* was computed from the slope log σ_dc_ vs. 1/*T* by using the following expression [47,48]:(3)$$\sigma_{dc}=\sigma_{0}exp\left(\frac{-E_{dc}}{k_{b}T}\right).$$

According to Figure 8, there is no clear trend in the *E_dc_* values (vs. Y^3+^ addition), which corresponds to data in Figure 5a. The lack of an *E_dc_* trend indicates that there is no apparent loss in the ionic component of CD conductivity which can be assigned to the different content of the Y-TZP samples. However, the *E_dc_* values presented in Figure 8 correspond well to values obtained for, e.g., different lithium borate glasses [47,48].

If the aforementioned quick analyses are taken into consideration, then the EEC values (*R*_1_ and *Y*_0,2_) can be utilized to monitor Y^3+^ (i.e., oxygen vacancies) content in the grains and grain boundaries of the Y-TZP materials studied in this work.

## 4. Conclusions

In this study, the effect of Y^3+^ addition on different yttria-stabilized tetragonal zirconia polycrystal (Y-TZP) samples was investigated. The Y-TZP samples were prepared as discs by using CAM/CAD technology. The morphology of the discs and their structural and electrical properties were studied by diverse techniques.

It was demonstrated that a higher Y^3+^ content increased the grain size and yielded a denser Y-TZP sample morphology. The analyses showed that the samples with the lowest and highest [Y^3+^] were characterized by the smallest and highest grain size, respectively.

Furthermore, an increase in the Y^3+^ content enlarged the fraction of the cubic phase of the Y-TZP samples. Herein, the detailed Rietveld analysis showed that Y^3+^ ions mainly incorporated the tetragonal phase when the ion concentration was higher.

The investigation of EEC parameter trends revealed that an increase in Y^3+^ content decreased the conductivity of the grains and increased the pseudocapacitance of the grain boundary. It was explained that Y^3+^ ions that were trapped within the grain boundary were responsible for the observed EEC parameter trends.

Overall, several findings from this work should be emphasized. First, it should be stressed that the grain size in the investigated Y-TZP materials was governed by Y^3+^ addition. Second, holes were only detected in the disc with the lowest grain density, which implies that densification promotes the materials’ endurance in CAM/CAD processing. Third, according to the Rietveld analysis, the content of the cubic phase was increased with Y^3+^ ion addition, but then these ions mainly incorporated the tetragonal phase. Finally, the results in this work suggest that the *R*_1_ value can be used to monitor the Y^3+^ (i.e., oxygen ion vacancy) content within the grains, whilst the Y_0,2_ value (i.e., pseudocapacitance) can be used to observe changes in both the grain boundary thickness and Y^3+^ content.

## Data Availability

Data sharing is not applicable to this article.

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
