# Peer review of "The Effect of Y3+ Addition on Morphology, Structure, and Electrical Properties of Yttria-Stabilized Tetragonal Zirconia Dental Materials"

_materials, 2022, doi:10.3390/ma15051800_

Round 1

Reviewer 1 Report

Please see comments in pdf

Author Response

Response to Reviewer #1,

Thank you for your comments and suggestions.

Q: The aim of the study is not described precise enough. The aims should be clearly structured and prioritized.

A: The aim of this study is rewritten.

Q: Table 1 is not self-explaining. I suggest a more intuitive design of the Table 

A: We have provided more data in Table 1.

Q: You mean in implant dentistry to roughen the part of the surface in contact with bone – right? But crowns are also a dental application and are not sandblasted. Please be more precise with this claim.

 A: To clarify, the crowns are generally sandblasted in the dental laboratory for cleaning purposes.  

Q: Figure 1 … content is defined with the smallest/greatest grain size Should be replaced by contend points out with the smallest/biggest grain size.

A: The Figure caption was modified

Q: should or will be?

A: This issue was resolved.

Q: funding needs to be completed.

A: This issue was resolved.

Reviewer 2 Report

The manuscript “The effect of Y3+ adding on morphology, structure, and electrical properties of yttria-stabilized tetragonal zirconia dental materials” reporting the impact of Y3+ doping on the morphology, structure, and electrical properties of the prepared Y-TZP samples. The disc-shaped samples were prepared by using CAM/CAD technology. The manuscript is well written and results are constrictive. This work could considered for publication and prove to be more interesting if the authors made the following minor changes/modifications.

  • Revised the abstract part, the statement are not meaningful i.e. “The aim of this work was to investigate the impact of various Y3+ addings on the morphology, structure, and electrical properties of the prepared Y-TZP samples” and should be brief including novelty of work.
  • Author should need to brief the problems and motivation behind this work in introduction section.
  • Activation energy is an important parameter to understand the electrical conduction phenomena, calculate it and compare it with published articles.
  • The manuscript can be improved by including the glass study such as “Solid State Ionics Volume 368, 1 October 2021, 115704 (https://doi.org/10.1016/j.ssi.2021.115704), and Materials Today: Proceedings Volume 45, Part 3, 2021, Pages 3722-3725; (https://doi.org/10.1016/j.matpr.2020.12.688).
  • Enlarge the font size in all figure.
  • Conclusion should be what you have concluded from this study rather than findings of every characterization technique. Rewrite it as it looks vague in original version.
  • Language needs improvement throughout the manuscript.

Author Response

Response to Reviewer #2,

Thank you for your comments and suggestions.

Q: The manuscript “The effect of Y3+ adding on morphology, structure, and electrical properties of yttria-stabilized tetragonal zirconia dental materials” reporting the impact of Y3+ doping on the morphology, structure, and electrical properties of the prepared Y-TZP samples. The disc-shaped samples were prepared by using CAM/CAD technology. The manuscript is well written and results are constrictive. This work could considered for publication and prove to be more interesting if the authors made the following minor changes/modifications.

A: Thank you for your comments and suggestions.

Q: Revised the abstract part, the statement are not meaningful i.e. “The aim of this work was to investigate the impact of various Y3+ addings on the morphology, structure, and electrical properties of the prepared Y-TZP samples” and should be brief including novelty of work.

A: The abstract was revised according to suggestions.

Q: Author should need to brief the problems and motivation behind this work in introduction section.

A: The Introduction part was modified according to the instructions.

Q: Activation energy is an important parameter to understand the electrical conduction phenomena, calculate it and compare it with published articles.The manuscript can be improved by including the glass study such as “Solid State Ionics Volume 368, 1 October 2021, 115704 (https://doi.org/10.1016/j.ssi.2021.115704), and Materials Today: Proceedings Volume 45, Part 3, 2021, Pages 3722-3725; (https://doi.org/10.1016/j.matpr.2020.12.688).

A: We have computed activation energy and we have cited the following paper: https://doi.org/10.1016/j.ssi.2021.115704

Q: Enlarge the font size in all figure.

A: Fonts were enlarged.

Q: Conclusion should be what you have concluded from this study rather than findings of every characterization technique. Rewrite it as it looks vague in original version.

A: Conclusion was modified according to the suggestions.

Q: Language needs improvement throughout the manuscript.

A: The language was rechecked.

Round 2

Reviewer 2 Report

Authors have satisfatory responses to raised queries. The work may accepted for publication.